# Selenomethionine Mitigates Effects of *Nocardia cyriacigeorgica*-Induced Inflammation, Oxidative Stress, and Apoptosis in Bovine Mammary Epithelial Cells

**DOI:** 10.3390/ijms252010976

**Published:** 2024-10-12

**Authors:** Talgat Assabayev, Jinge Han, Halihaxi Bahetijiang, Venera Abdrassilova, Muhammad Asfandyar Khan, Herman W. Barkema, Gang Liu, John P. Kastelic, Xueying Zhou, Bo Han

**Affiliations:** 1College of Veterinary Medicine, China Agricultural University, Beijing 100193, China; talgata@cau.edu.cn (T.A.); halihaxi@cau.edu.cn (H.B.); asfand@cau.edu.cn (M.A.K.); gangliu@cau.edu.cn (G.L.); 2College of Animal Science and Veterinary Medicine, Tianjin Agricultural University, Tianjin 300384, China; 2203020138@stu.tjau.edu.cn; 3Department of Normal Physiology with Biophysics Course, Asfendiyarov Kazakh National Medical University, Almaty 050012, Kazakhstan; abdrasilova.v@kaznmu.kz; 4Faculty of Veterinary Medicine, University of Calgary, Calgary, AB T2N 4N1, Canada; barkema@ucalgary.ca (H.W.B.); jpkastel@ucalgary.ca (J.P.K.)

**Keywords:** bovine mastitis, *Nocardia cyriacigeorgica*, selenomethionine, oxidative stress, apoptosis

## Abstract

*Nocardia cyriacigeorgica* causes bovine mastitis, reduces milk quantity and quality, and is often resistant to antimicrobials. Selenomethionine (SeMet) is a form of selenium, which reduces reactive oxygen species (ROS)-mediated apoptosis and intramammary infections. However, the protective effects of SeMet on *N. cyriacigeorgica*-infected bovine mammary epithelial cells (bMECs) are unclear. The objective of this study was to evaluate whether SeMet mitigated *N. cyriacigeorgica*-induced inflammatory injury, oxidative damage and apoptosis in bMECs. Cells were cultured with or without being pretreated with 40 µM of SeMet for 12 h, then challenged with *N. cyriacigeorgica* (multiplicity of infection = 5:1) for 6 h. Although *N. cyriacigeorgica* was resistant to lincomycin, erythromycin, enrofloxacin, penicillin, amoxicillin, cephalonium, cephalexin, and ceftriaxone, 40 μM SeMet increased cell viability and inhibited lactate dehydrogenase release in infected bMECs. Furthermore, *N. cyriacigeorgica* significantly induced mRNA production and protein expression of TNF-α, IL-1β, IL-6, and IL-8 at 6 h. Cell membrane rupture, cristae degeneration and mitochondria swelling were evident with transmission electron microscopy. Superoxide dismutase (SOD) and glutathione peroxidase (GSH-px) activities were down-regulated after 3, 6, or 12 h, whereas malondialdehyde (MDA) and ROS contents were significantly upregulated, with cell damage and apoptosis rapidly evident (the latter increased significantly in a time-dependent manner). In contrast, bMECs pretreated with 40 μM SeMet before infection, SOD, and GSH-px activities were upregulated (*p* < 0.05); MDA and ROS concentrations were downregulated (*p* < 0.05), and apoptosis was reduced (*p* < 0.05). In conclusion, 40 μM SeMet alleviated inflammation, oxidative stress and apoptosis induced by *N. cyriacigeorgica* in bMECs *cultured* in vitro.

## 1. Introduction

Bovine mastitis affects dairy cows worldwide, with Nocardia spp. often causing mastitis in dairy cows, and enormous economic losses [1]. Furthermore, Nocardia can cause various local and systemic infections in humans and animals, and is transmitted in a variety of ways, including inhalation, ingestion, or traumatic implantation, with dissemination via lymphatics and blood [2]. *Nocardia cyriacigeorgica* causes chronic inflammation that does not respond to therapy [3] and is the species of Nocardia most commonly linked to bovine mastitis [4,5,6]. Nocardia mastitis causes granulomatous inflammation in cattle, which is initially acute and eventually becomes chronic [3,7].

As Nocardia spp. are resistant to many antimicrobials commonly used in veterinary medicine, mastitis caused by these bacteria results in chronic intramammary infection, mammary dysfunction, and cow culling [8]. Longstanding antibiotic use has increased antimicrobial resistance in *N. cyriacigeorgica*, creating global concerns for human and animal health [9]. However, antimicrobial resistance data on isolates associated with mastitis are limited.

Bacterial mastitis causes oxidative stress by promoting production of reactive oxygen species (ROS). Malondialdehyde (MDA) is frequently used to assess lipid peroxidation [10], which is counteracted by specific enzymes [11]. Oxidative stress can damage udder tissues and decrease milk production [12]. With cells cultured in vitro, damaged membranes release intracellular lactate dehydrogenase (LDH), which is an indicator of cytopathic effects [13]. In addition, ROS generated during oxidative stress could result in cytotoxic effects through protein and amino acid oxidation, DNA damage, altered intracellular signaling, inhibition of antioxidant defense systems, and tissue damage.

Cell death and inflammation are closely linked during mastitis due to *Staphylococcus aureus* [14]. Pyroptosis, a programmed necrosis triggered by the gasdermin protein family, often occurs after inflammatory caspase activation. Many pathogens invade host cells and activate cell-intrinsic death mechanisms, including pyroptosis, apoptosis, and necroptosis. Apoptosis occurs in response to various stimuli, including infection and injury [15]. Apoptosis has characteristic ultrastructural changes, including cell shrinkage, chromatin condensation, nuclear fragmentation, and apoptotic bodies [16]. The role of apoptosis in the context of infection or injury, particularly in bovine mastitis caused by *N. cyriacigeorgica*, is an area of ongoing research [3,16]. Although apoptosis has a role in host defense mechanisms against microbial invasion, specific interactions between *N. cyriacigeorgica* and bMECs regarding apoptosis are not well studied. Investigations in bovine mastitis caused by *N. cyriacigeorgica* have primarily focused on prevalence, molecular characterization, immunogenic proteins, and virulence determinants [3,15,16]. Further research on the apoptotic response of bMECs to *N. cyriacigeorgica* infection may provide insights into host–pathogen interactions and pathogenesis.

Selenomethionine (SeMet) is an important micronutrient that can prevent mastitis in dairy cows through various biological pathways, including suppressing apoptosis [17]. Furthermore, selenium deficiency increases susceptibility to various diseases and decreases productivity globally [18]. Nevertheless, the effects of SeMet on apoptosis in *N. cyriacigeorgica*-infected bMECs are unknown. The objective of this study was to evaluate whether SeMet mitigated *N. cyriacigeorgica*-induced inflammatory injury, oxidative damage, and apoptosis in bMECs.

## 2. Results

### 2.1. Antimicrobial Susceptibility

The *N. cyriacigeorgica* was resistant to most antibiotics tested, although it was susceptible to gentamycin (66.7%) and ceftiofur (93.3%) (Table 1).

### 2.2. Cytotoxicity Effects of SeMet on bMECs

Based on a CCK-8 assay, bMECs treated with 40 µM of SeMet had higher viability (*p* < 0.01; Figure 1A) compared to the control group. However, 60 or 80 µM of SeMet significantly damaged bMECs at 12 h. In bMECs infected with *N. cyriacigeorgica*, 40 µM SeMet increased cell viability (*p* < 0.05; Figure 1B). Although 40 µM SeMet inhibited LDH release, > 60 μM SeMet increased LDH release (*p* < 0.01; Figure 1C). Furthermore, 40 µM SeMet reduced LDH release (*p* < 0.01) in bMECs infected with *N. cyriacigeorgica* (Figure 1D) without cytotoxicity.

### 2.3. Transmission Electron Microscopy (TEM) of bMECs

In bMECs, at 6 h post-infection with *N. cyriacigeorgica*, ultrastructure changes included mitochondria and endoplasmic reticulum swelling, organelle loss, cell membrane rupture, perinuclear space, and cellular dysfunction (Figure 2). Moreover, Nocardia were in the nucleus and cytoplasm, with indications that this bacterium caused apoptosis. In contrast, the SeMet-treated group had fewer intracellular bacteria, and cell morphology was similar to the untreated group.

### 2.4. Gene Expression and Production of Pro-Inflammatory Cytokines

Compared to the control group, both mRNA and protein expression levels of TNF-α, IL-1β, IL-6, and IL-8 in bMECs were downregulated and decreased in cells pretreated with 40 μM of SeMet and then infected with *N. cyriacigeorgica* (Figure 3, *p* < 0.01; Figure 4, *p* < 0.05).

### 2.5. Superoxide Dismutase (SOD) and Glutathione Peroxidase (GSH-px) Activities and MDA Concentration

The activities of antioxidant enzymes SOD and GSH-px were decreased (*p* < 0.01), whereas MDA concentration was increased (*p* < 0.01) in infected cells versus controls. However, in cells pretreated with 40 μM SeMet and later challenged with *N. cyriacigeorgica*, activities of both SOD and GSH-px increased (*p* < 0.05), whereas MDA content decreased (*p* < 0.01) compared to the infection group (Figure 5).

### 2.6. N. Cyriacigeorgica Stimulated Reactive Oxygen Species (ROS)

At 6 h after *N. cyriacigeorgica* infection, ROS concentrations in bMECs increased (*p* < 0.01) compared to the control group. However, in cells pretreated with 40 μM SeMet, ROS concentrations were decreased (*p* < 0.01) compared to the infection group. Furthermore, ROS concentration was not different between the group pretreated with SeMet and uninfected bMECs (*p* > 0.05, Figure 6).

### 2.7. N. cyriacigeorgica Induced Apoptosis in bMECs

The percentage of apoptosis in bMECs infected with *N. cyriacigeorgica* was increased in a time-dependent manner. At 3 h, apoptosis of the infection group was higher than the control group (*p* < 0.05, Figure 7). In addition, apoptosis in the infection group was higher (*p* < 0.01) than in the SeMet-treated group.

## 3. Discussion

Outbreaks of bovine mastitis caused by *Nocardia* spp. are common worldwide, including in China [3]. Key factors contributing to *Nocardia*-induced bovine mastitis include poor management and hygiene, such as inadequate disinfection of teats before and after milking, lack of proper sanitation, and use of contaminated drugs or containers for treatments [19]. Additionally, factors such as decreased immunity due to high milk production, calving, and other diseases can increase the susceptibility of cows to *Nocardia* infection.

As *N. cyriacigeorgica* adheres to and penetrate bMECs, causing cell damage and apoptosis [20], it is necessary to seek effective treatments. Although *Nocardia* spp. have diverse sensitivity to antibiotics, which can be used for phenotypic identification [21], strains affecting cattle are usually resistant to common veterinary antimicrobials. Antimicrobial susceptibility (MIC method) of *N. cyriacigeorgica* to 10 antimicrobials was determined. The *N. cyriacigeorgica* were susceptible only to gentamycin and ceftiofur, but were resistant to eight other microbials. *N. puris, N. veterana, N. cyriacigeorgica, N. arthritidis,* and *N. africana* were resistant to cloxacillin (75%), ampicillin (55%), and cefoperazone (47%), but were less resistant to amikacin, cefuroxime, or gentamicin [7]. However, there are limited reports characterizing antimicrobial sensitivity/resistance of *Nocardia* spp. from bovine mastitis [22]. Moreover, based on the intracellular location of the pathogen and virulence characteristics that cause a pyogranulomatous reaction, treatment of *Nocardia* mastitis is usually not curative. Therefore, early identification of clinical and subclinical cases of bovine mastitis by microbiological identification and drug sensitivity tests in vitro is crucial to improve response to treatment or promote timely culling [23]. Although ceftriaxone and imipenem were used for the treatment of moderate and severe nocardiosis in Australia, only 58.8% of isolates were sensitive to ceftriaxone and 41.0% to imipenem [2]. A similar susceptibility pattern was reported in other Australian studies for ceftriaxone (44–60%) and imipenem (33.7–48%), based on a bone mineral density test [8,24]. Therefore, SeMet was assessed as a potential alternative treatment method for *Nocardia* mastitis in the current research.

In this study, the effects of various concentrations of SeMet on cell viability, LDH assay, antioxidant status and oxidative stress in bMECs were determined. Although 40 µM SeMet had protective effects on bMECs, higher concentrations (i.e., 60 or 80 µM) were toxic. It was reported that SeMet alleviated ESBL *E. coli*-induced inflammation by activating the selenoprotein S-mediated TLR4/NF-κB signaling pathway in bMECs [25], which is consistent with the present results. Furthermore, SeMet enhanced expression of glutathione peroxidase 1 and 3 both in vivo and in vitro, and improved the growth and viability of bMECs [26]; whereas selenium concentrations from 50 to 100 nmol/L improved antioxidant function but did not affect milk protein synthesis in healthy bMECs. In addition, selenium mitigated lipopolysaccharide-induced damage by increasing antioxidant markers and stimulating milk protein synthesis and expression of milk protein-related genes in bMECs [27].

Oxidative stress impacts various physiological cell functions, including the immune response and the reproductive system [28,29]. Selenomethionine is regarded as an important element of the antioxidant system, as selenium enhances bMECs’ antioxidant function and protects cells from diethylenetriamine/NO-induced oxidative damage, mainly by increasing TrxR activity and reducing NO concentration via modulation of Nrf2 and MAPK signalling pathways [30]. Dietary intake of selenomethionine is directly correlated with the activities of various antioxidant enzymes (GSH-px, SOD, MDA, and ROS). In this study, SOD and GSH-px activities were downregulated by *N. cyriacigeorgica* infection. However, pretreatment with 40 µm SeMet upregulated SOD and GSH-px activities. Furthermore, in bMECs, MDA is considered a major product of lipid peroxidation and a marker of oxidative stress and antioxidant status [29]. In this study, whereas MDA and ROS contents were upregulated by *N. cyriacigeorgica* infection, SeMet pretreatment downregulated MDA and ROS, implying SeMet had a protective effect. Similarly, dairy cows given selenium yeast supplements (SeMet is the predominant form of selenium) had increased serum GSH-px activity compared to cows given sodium selenite supplements [31]. Furthermore, 0.3 mg/kg DM of hydroxy-selenomethionine enhanced GSH-px activity in dairy cow serum compared to the same dose of sodium selenite [32]. In studies using primary rainbow trout hepatocytes to determine the effects of SeMet and sodium selenite on arsenite cytotoxicity, SeMet attenuated oxidative stress by increasing SOD [32,33]. Furthermore, SeMet increased the capacity of bMECs to withstand oxidative stress, as indicated by decreased ROS and MDA [32], which is consistent with current results. Additionally, selenium suppressed *S. aureus*-induced inflammation by inhibiting the NLRP3 inflammasome and ROS activation in bMECs [34].

The current study aimed to determine whether SeMet mitigated inflammatory and oxidative damage in bMECs caused by *N. cyriacigeorgica* strains isolated from mastitic milk. Expression of TNF-α, IL-1β, IL-6, and IL-8 were significantly upregulated in bMECs infected by *N. cyriacigeorgica*. In contrast, pretreatment with 40 µM SeMet significantly downregulated gene expression of TNF-α, IL-1β, IL-6, and IL-8. Therefore, SeMet reduced production of inflammatory cytokines and decreased inflammatory damage to bMECs induced by *N. cyriacigeorgica*. Similarly, SeMet increased the autophagic flux and inhibited the NF-κB-mediated inflammatory response of bMECs caused by *Klebsiella pneumoniae* [35]. Furthermore, gene expression of TNF-α, IL-1β, IL-6, and IL-8 were also significantly upregulated 1 to 5 h after infection by *K. pneumoniae* [36], although SeMet reduced inflammation and protein expression of TLR4, Ikappa-B, NF-κB, and TNF-α in bMECs and macrophages infected with ESBL *E. coli* [17]. Increased selenium concentrations decreased expression of IL-1β and IL-6 [37]. In addition, selenium reduced gene expression of inflammatory mediators TNF-α, IL-1β, and IL-6 through the TLR2, NF-κB, and MAPK signaling pathway in *S. aureus-*infected bMECs, implying this was the basis of the anti-inflammatory effect of selenium [38]. Pre-treatment with various concentrations of SeMet significantly downregulated protein expression in ESBL-producing *E. coli*-infected bMECs [17], which is consistent with current findings. IL-1β triggers neutrophil migration and its expression level is related to severity of inflammation. Moreover, IL-6 promotes cell proliferation and differentiation and induces acute gene expression in hepatocytes, increasing inflammation [39].

In the current study, SeMet clearly had a protective effect against apoptosis induced by *N. cyriacigeorgica* infection of bMECs. Although apoptosis in the *N. cyriacigeorgica* infection group was significantly upregulated in a time-dependent manner, this was prevented by pretreatment with 40 µM SeMet, indicating that SeMet alleviated *N. cyriacigeorgica*-induced cell apoptosis, as this bacterium induces both apoptotic and necrotic changes in bMECs after 6 h [20]. Based on transcription levels of apoptosis-associated genes, overexpression of heme-binding protein could induce cell apoptosis [40]. MTSP3141, a secretory protein from *N. seriolae*, can target mitochondria and induce apoptosis in host cells, and function as a virulence factor [41]. However, selenium-related treatments protected against apoptosis, whereas selenium-enriched plant supplements significantly downregulated apoptosis of ovarian cells [42]. SeMet supplementation protected bMECs from H_2_O_2_-induced apoptosis and enhanced cell proliferation and viability under oxidative stress [43]. Furthermore, 500 nM of Na_2_ SeO_3_ or 30 μM of L-selenium-methylselenocysteine significantly suppressed apoptosis and reduced mitochondrial membrane potential in murine liver cells, whereas 0.45 mg/kg dietary selenium also remarkably ameliorated liver damage in mice fed high-fat diets [44]. Extracellular vesicles derived from selenium-deficient bMECs induced oxidative stress, provoking apoptosis and inflammation via endoplasmic reticulum stress and PI3K/Akt/mTOR signaling pathway, contributing to explain susceptibility to mastitis due to selenium deficiency [45]. SeMet decreased apoptosis by increasing the overall antioxidant capacity and reducing cellular ROS [46], which is consistent with current results.

## 4. Materials and Methods

### 4.1. Cell Culture

A bovine mammary epithelial cell line (MAC-T) (Shanghai Jingma Biological Technology Co., Ltd., Minhang district, Shanghai, China) was cultured in Dulbecco’s modified Eagle medium (DMEM) containing 10% fetal bovine serum and 1% antibiotic mix (100 μg/mL streptomycin and 100 U/mL penicillin), in an incubator with 5% CO_2_ at 37 °C. At 70–80% confluence, cells were detached and seeded into 6- or 96-well plates for assays (Figure 8).

### 4.2. Bacterial Culture and Basic Tests

*N. cyriacigeorgica,* previously isolated in our laboratory from milk samples with bovine mastitis, was activated from frozen stocks by culturing on tryptose soya agar (DifcoTM, Becton Dickison, Sparks, MD, USA) supplemented with 5% defibrinated sheep blood. It was maintained in aerobic conditions at 37 °C for 72 h, then sub-cultured in Brain Heart Infusion (BHI) broth for the following experiments. The detailed procedure was the same as described by Chen (2017) [3].

### 4.3. Minimum Inhibitory Concentration (MIC) Determination

The antimicrobial susceptibility of *N. cyriacigeorgica* was assessed with a broth microdilution assay, recommended by the Clinical and Laboratory Standards Institute (CLSI, 2017) [47]. *Staphylococcus aureus* ATCC 29213 was used for quality control [48]. The MIC was the lowest concentration of antimicrobial that totally inhibited bacterial growth. For MIC analyses, 96-well microplates were used, with 200 µL of BHI and antimicrobial added. After incubation of *N. cyriacigeorgica* in BHI broth for 48 h, MIC values were the lowest concentrations of the antimicrobial without visible bacterial growth. We tested the following antimicrobial agents: gentamycin, lincomycin, erythromycin, enrofloxacin, penicillin, amoxicillin, cephalonium, cephalexin, ceftriaxone, and ceftiofur; with MIC values based on visual observation. The concentrations needed to inhibit visible growth of 50% and 90% of isolates were defined as MIC_50_ and MIC_90_, respectively.

### 4.4. Assessment of Cell Viability by CCK-8

Cell Counting Kit-8 (CCK-8 kit; Beyotime Institute of Biotechnology, Beijing, China) is a rapid and highly sensitive method for measuring cell proliferation and cytotoxicity. MAC-T cells (~5000 cells/well) were plated in 96-well plates with 0, 5, 10, 20, 40, 60, or 80 µmol/L SeMet for 12 h. After treatment, cells were infected with *N. cyriacigeorgica* (MOI = 5:1) and incubated at 37 °C for 6 h and a CCK-8 kit used to assess cell viability, in accordance with the manufacturer’s directions. After 10 μL of CCK-8 solution was added in each well, cells were incubated at 37 °C for 2 h. Absorbance was recorded at 450 nm in a DNM 9602 microplate reader. Cell viability was evaluated as a percentage (%) relative to the control.

### 4.5. Assessment of Lactate Dehydrogenase (LDH) Release

Cytotoxic effects of *N. cyriacigeorgica* on bMECs were evaluated using an LDH assay kit (Beyotime Institute of Biotechnology). The bMECs cells were cultured at 37 °C with 5% CO_2_ in 96-well plates and incubated with 0, 5, 10, 20, 40, 60, or 80 µmol/L SeMet for 12 h. Afterwards, treated cells were infected with *N. cyriacigeorgica* at a MOI of 5:1 for 4 or 6 h. After incubation, the cell culture plate was centrifuged at 400× *g* for 5 min, and then 120 µL of supernatant was transferred to a new 96-well plate, with 60 µL of reaction mixture added to each well, followed by incubation at room temperature in the dark for 30 min on a horizontal shaker (150 rpm). Optical density (OD) was measured by a spectrophotometer at 490 nm.

### 4.6. Transmission Electron Microscopy (TEM) of BMECs

For TEM, bMECs were seeded on 6-well plates and infected with *N. cyriacigeorgica* (MOI = 5:1) for 6 h. After infection, cells were collected with a cell scraper and centrifuged at 1000× *g* for 5 min. They were washed twice with PBS, fixed in 2.5% glutaraldehyde for 2 h and 1% osmium tetroxide for 2 h at 4 °C, dehydrated in a graded ethanol series, embedded in epoxy resin acetone mixtures (2:1) for 2 h, then immersed in pure resin for 12 h at 37 °C. After polymerization, ultrathin sections (50~70 nm) were cut, stained with 1% uranyl acetate and lead citrate, and examined with transmission electron microscopy (JEM-1400, Tokyo, Japan) at 200 KeV.

### 4.7. Determination of Pro-Inflammatory Cytokines

BMECs were infected with *N. cyriacigeorgica* (MOI = 5:1) for 4 h and remained in culture medium with antibiotics and lysozyme. The supernatant was collected by centrifugation at 12,000× *g* for 10 min at 4 °C. An ELISA kit (Jiangsu Meimian Industrial Co., Ltd., Yancheng, China) with a detection range of 0.1 to 20.0 ng/mL was used (according to manufacturer’s instructions) to assess concentrations of pro-inflammatory cytokines TNF-α, IL-1β, IL-6, and IL-8 in cell supernatant. Absorbance at 450 nm was measured on a DNM 9602 microplate reader. All measurements were done 3 times, and the mean of 3 independent readings used for statistical analyses.

### 4.8. RNA Extraction and Reverse Transcriptase Quantitative Polymerase Chain Reaction (RT-qPCR)

Briefly, bMECs were washed once with PBS and harvested with 1 mL of TransZol Up lysis solution to extract total RNA, according to label instructions (TransGen Biotech Co., Ltd., Beijing, China). The cDNA was synthesized by TransScript^®^ II All-in-One First-Strand cDNA Synthesis SuperMix for qPCR (One-Step gDNA Removal; TransGen Biotech). Concentrations of total RNA and cDNA were determined by absorbance at 260 and 280 nm on a NanoDrop-2000 spectrophotometer (Thermo Fisher Scientific, Waltham, MA, USA). A 20-µL RT-qPCR mixture was used, comprised of 10 µL of 2 × RealStar Fast SYBR qPCR mix (GenStar, Beijing, China), 10 µM (0.5 µL) of each primer, 1 µL cDNA, and 8 µL of nuclease-free water. Expression levels of mRNA were tested by real-time PCR as follows: predenaturation at 94 °C for 2 min; 40 cycles of denaturation at 95 °C for 5 s; and annealing at 60 °C for 30 s using an Applied Biosystems StepOnePlus Real Time PCR system (Thermo Fisher Scientific). Primers for a housekeeping gene (GAPDH) and genes encoding TNF-α, IL-1β, IL-6, and IL-8 were used (Table 2). Cycle threshold (Ct) values and 2^−ΔΔCt^ values were calculated to determine expression.

### 4.9. Determination of MDA Concentration, SOD and GSH-px Activities

The bMECs were cultured in 6-well plates and incubated with *N. cyriacigeorgica* (MOI 0 and 5) for 6 h. After infection, bMECs were washed 3 times with ice-cold PBS, lysed in SOD sample preparation solution, then centrifuged at 12,000× *g* at 4 °C for 3–5 min and the supernatant was collected for a subsequent assay. Then, MDA concentrations and activities of GSH-px and SOD were detected using Cell MDA, GSH-px and SOD assay kits (Cat. no. S0131, S0056 and S0101, respectively; Beyotime Biotechnology, Shanghai, China), according to the manufacturer’s instructions. Readings were recorded with a DNM 9602 microplate reader at 450 and 340 nm. Each group was assayed 30 times (10 replicates per group and each sample assayed in triplicate).

### 4.10. Measurement of Reactive Oxygen Species (ROS) Concentrations

Intracellular ROS were measured by the reactive oxygen species assay kit (Beyotime Institute of Biotechnology, Shanghai, China) with an oxidation-sensitive fluorescent probe (DCFH-DA). Cells were seeded into 6-well plates with glass coverslips, treated, then cells were washed once using PBS and stained in darkness with 100 μM of diluted DCFH-DA for 20 min at 37 °C and then washed 3 times with DMEM. The DCFH-DA was diluted 1000 times with serum-free DMEM/F12 culture medium, according to the manufacturer’s instructions. Finally, cells were stained with DAPI for 10 min. After washing 3 times in ice-cold PBS, samples were viewed with a laser confocal microscope (NIKON A1 HD25, Tokyo, Japan).

### 4.11. Determination of Apoptosis

Apoptosis induced by *N. cyriacigeorgica* in bMECs was determined with an Annexin V-FITC/propidium iodide (PI) analysis kit (Yeasen Biotechnology (Shanghai) Co., Ltd., Shanghai, China) according to the manufacturer’s instructions. Cells were plated into 6-well plates and infected with *N. cyriacigeorgica* at MOI of 0 or 5 for 3, 6, 9, or 12 h. After infection, cell culture medium was discarded and cells were harvested with 0.25% trypsin without EDTA and collected by pipetting into 1.5 mL PCR tubes. Cells were centrifuged at 1000× *g* 4 °C for 5 min, washed twice with ice-cold PBS, re-suspended in 100 μL of 1 × binding buffer, and stained with 5 μL of Annexin V (FITC) and 10 μL of PI. After 15 min in the dark at room temperature, cells were assessed (FAC Scan Flow Cytometer; Becton, Dickinson and Co., East Rutherford, NJ, USA), as follows: Q1 = represents necrotic cells; Q2 = late apoptotic cells; Q3 = early apoptotic cells; and Q4 = late live cells or normal cells.

### 4.12. Statistical Analyses

Results were expressed as the mean ± standard deviation (SD) from at least 3 independent experiments. GraphPad Prism 8.0 (GraphPad Software, Inc., San Diego, CA, USA) was used and significance was assessed using SPSS 26.0 (IBM Corp., Armonk, NY, USA). Statistical differences were assessed with one-way analysis of variance (ANOVA), followed by the Tukey’s multiple comparison post hoc test. Significance was accepted at * *p* < 0.05; ** *p* < 0.01.

## 5. Conclusions

In conclusion, *Nocardia cyriacigeorgica* infection of bMECs induced structural disruption and mitochondrial degeneration. However, pretreatment with 40 µM SeMet had the following protective actions: upregulated antioxidant enzyme activities, improved mRNA expressions and protein synthesis rates, reduced oxidative stress and apoptosis, and protected against inflammatory damage. An optimal dose of selenomethionine had cytoprotective roles in bMECs, although higher doses were cytotoxic. These findings highlighted the potential therapeutic benefits of selenomethionine supplementation in mitigating oxidative stress, inflammation, apoptosis, and structural damage induced by bacterial infection in bMECs.

## Figures and Tables

**Figure 1 ijms-25-10976-f001:**
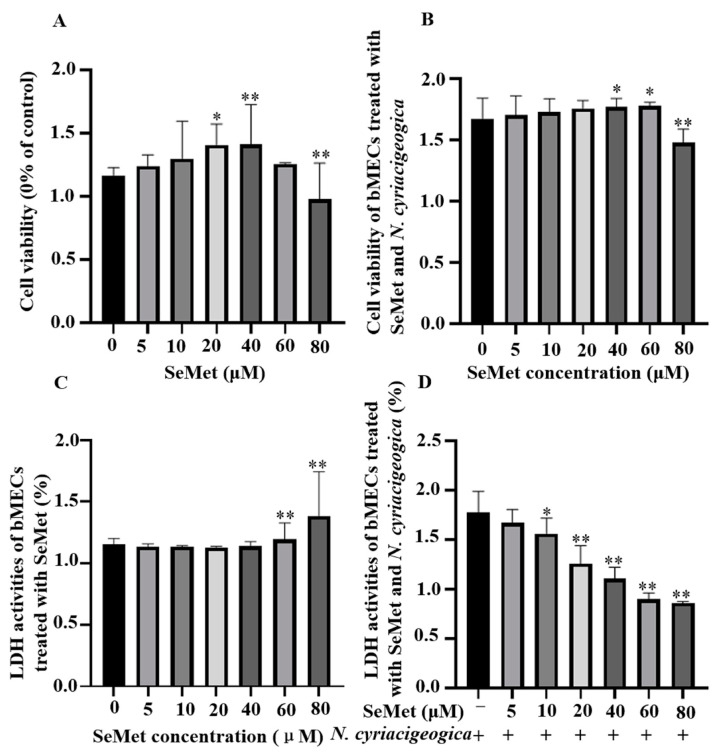
Effects of SeMet with or without *N. cyriacigeorgica* on viability of bMECs and cytotoxicity of *N. cyriacigeorgica*. Cell viability of bMECs exposed to various concentrations of SeMet for 12 h, followed by exposure to *N. cyriacigeorgica* for 6 h. (**A**) Cell viability was determined with a CCK-8 assay in SeMet-treated bMECs. (**B**) Cell viability was determined with a CCK-8 assay in SeMet-treated bMECs subsequently infected with *N. cyriacigeorgica*. (**C**) LDH release detected by an LDH assay kit in SeMet-treated bMECs. (**D**) LDH activities determined by an LDH assay kit in SeMet-treated bMECs that were subsequently infected with *N. cyriacigeorgica*. Data are means ± SD of 3 independent experiments. * *p* < 0.05; ** *p* < 0.01.

**Figure 2 ijms-25-10976-f002:**
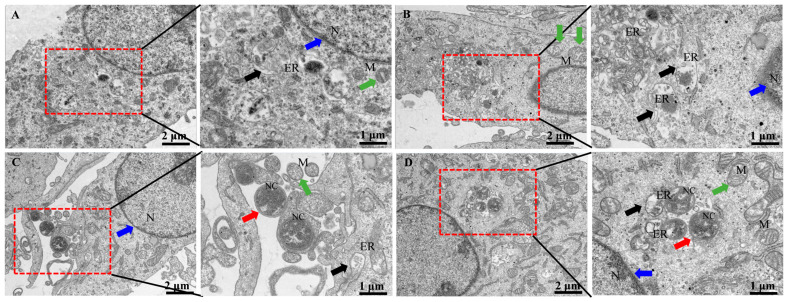
Ultrastructural changes of bMECs infected with *N. cyriacigeorgica* and examined with transmission electron microscopy. (**A**) Control group bMECs had abundant mitochondria (green arrow) in cytoplasm, and normal nucleus (blue arrow), and normal endoplasmic reticulum (black arrow); (**B**) in the SeMet-treated group, all bMECs components appeared unaltered; (**C**) *N. cyriacigeorgica* infection group, the red arrow indicates *N. cyriacigeorgica* (NC); note that the endoplasmic reticulum (black arrow) was swollen, indicating the cells were under stress. The green arrow indicates mitochondria, which appeared enlarged, with disruption of mitochondrial cristae (compared to the control group), whereas the blue arrow indicates the nucleus; (**D**) pretreatment with SeMet before *N. cyriacigeorgica* infection. The red arrow indicates *N. cyriacigeorgica* (NC), the green arrow indicates enlarged mitochondria, and black arrows indicate swollen endoplasmic reticulum which is under stress. Note the protective effects of SeMet on bMECs, including mitochondria (green arrow), endoplasmic reticulum (black arrow), and nucleus (blue arrow), with fewer changes compared to the infection group.

**Figure 3 ijms-25-10976-f003:**
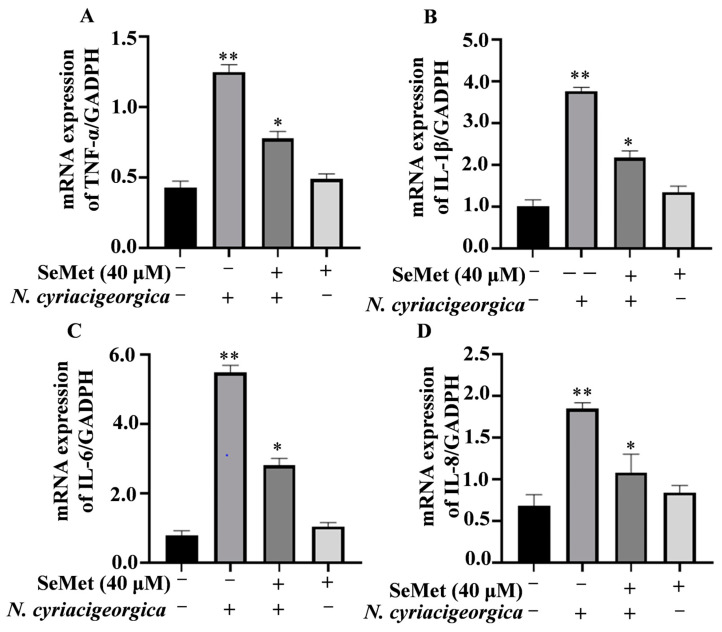
Effects of SeMet on gene expression of proinflammatory cytokines. Expression of: (**A**) TNF-α; (**B**) IL-1β; (**C**) IL-6; and (**D**) IL-8 in bMECs pretreated with 40 μM SeMet or nothing for 6 h before infection with *N. cyriacigeorgica* (multiplicity of infection (MOI) = 5:1) for 4 h. The mRNA expression of TNF-α, IL-1β, IL-6, and IL-8 were downregulated compared to the control group. Data are means ± SD of three independent experiments. * *p* < 0.05; ** *p* < 0.01.

**Figure 4 ijms-25-10976-f004:**
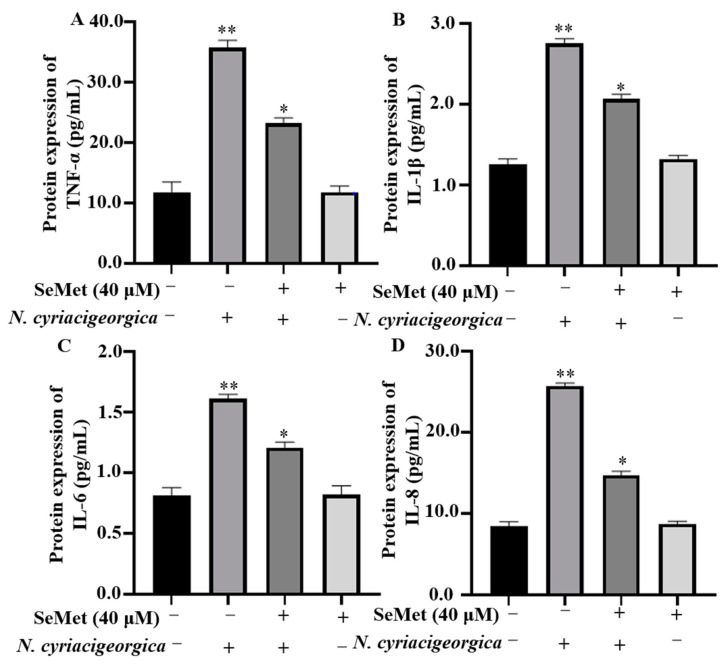
SeMet alleviated *N. cyriacigeorgica-*induced inflammatory responses in bMECs. The bMECs were treated with 40 μM of SeMet for 6 h and then infected with *N. cyriacigeorgica* for 4 h. SeMet pretreatment alleviated *N. cyriacigeorgica*-induced release of inflammatory cytokines in bMECs. (**A**) TNF-α; (**B**) IL-1β; (**C**) IL-6; and (**D**) IL-8. Data are means ± SD of three independent experiments. * *p* < 0.05; ** *p* < 0.01.

**Figure 5 ijms-25-10976-f005:**
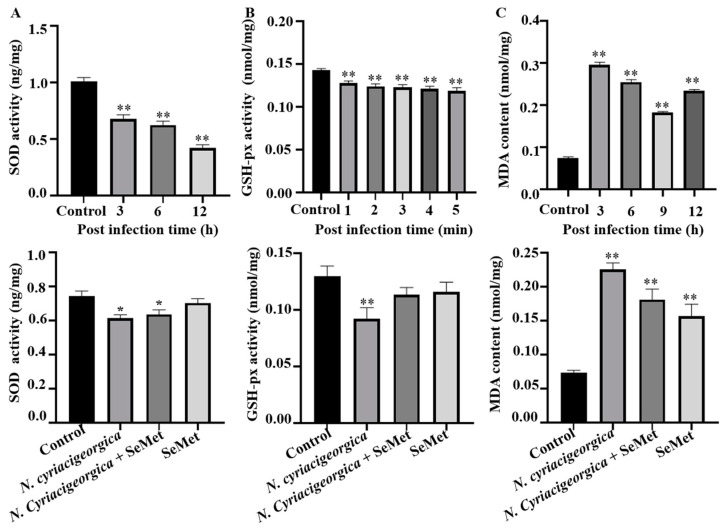
Effects of SeMet on *N. cyriacigeorgica* induced oxidative stress indicators in bMECs. (**A**) Superoxide dismutase (SOD) activities of bMECs infected *N. cyriacigeorgica* for 3, 6, or 12 h were decreased gradually (*p* < 0.01) but increased (*p* < 0.05) with 40 μM SeMet pretreament; (**B**) glutathione peroxidase GSH-px activity for 1, 2, 3, 4 or 5 min were decreased (*p* < 0.01) but increased (*p* < 0.05) with 40 μM SeMet pretreatment; (**C**) malondialdehyde (MDA) content in bMECs infected with *N. cyriacigeorgica* for 3, 6, 9, or 12 h were increased (*p* < 0.01), but were decreased (*p* < 0.05) with 40 μM SeMet pretreatment. Means ± SD of three independent experiments. Data are means ± SD of three independent experiments. * *p* < 0.05, ** *p* < 0.01.

**Figure 6 ijms-25-10976-f006:**
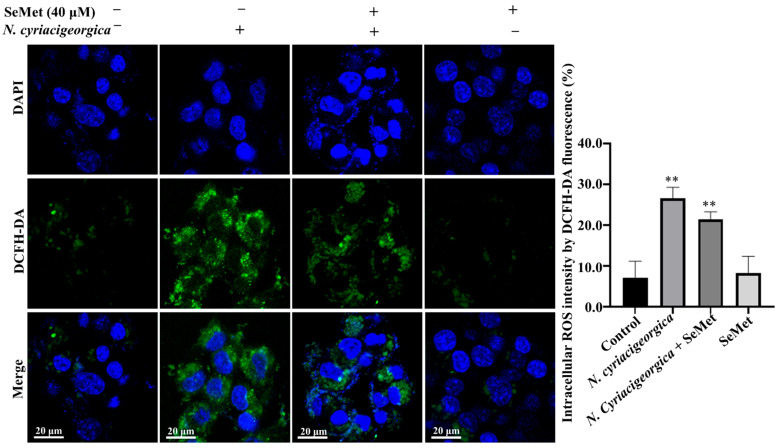
Effects of SeMet on *N. cyriacigeorgica*-induced increased in ROS concentrations in bMECs. In bMECs infected with *N. cyriacigeorgica*, ROS was increased (*p* < 0.01). However, in bMECs pretreated with 40 μM SeMet for 6 h before infection with *N. cyriacigeorgica* (MOI = 5:1), the ROS concentration was decreased (*p* < 0.01). In addition, ROS concentration of bMECs only pretreated with 40 μM SeMet for 6 h was not significantly different from the control. Data are means ± SD of three independent experiments. ** *p* < 0.01.

**Figure 7 ijms-25-10976-f007:**
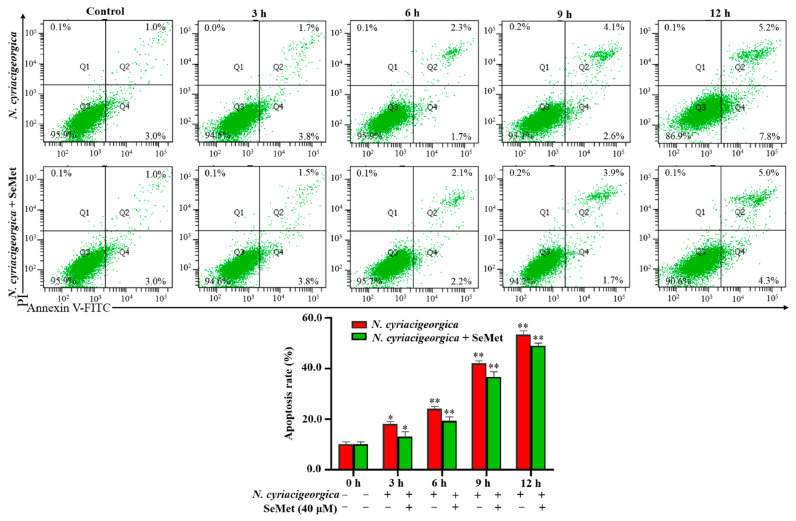
Apoptosis of bMECs treated with 40 μM SeMet for 12 h and subsequently infected with *N. cyriacigeorgica*. The bMECs were treated with 40 μM SeMet for 12 h and then infected with *N. cyriacigeorgica* at MOI of 0 and 5 for 3, 6, 9, or 12 h. The percentage of apoptosis in bMECs infected with *N. cyriacigeorgica* was increased in a time-dependent manner. Apoptosis of the infection group was higher (*p* < 0.01) at various time points compared to the group given 40 μM SeMet. Therefore, SeMet mitigated apoptosis induced by *N. cyriacigeorgica*. Data are means ± SD of three independent experiments. * *p* < 0.05; ** *p* < 0.01.

**Figure 8 ijms-25-10976-f008:**
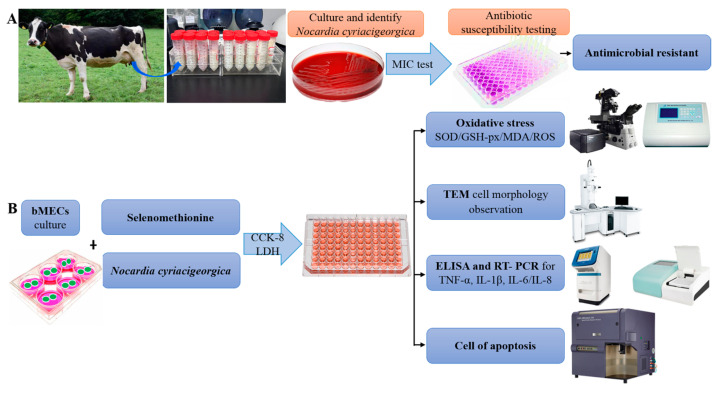
Schematic diagram of the experimental protocol. (**A**) Identification of bovine mastitis milk samples and antimicrobial resistance. (**B**) Selenomethionine alleviates effects of *Nocardia cyriacigeorgica-*induced inflammation, oxidative stress, and apoptosis in bovine mammary epithelial cells.

**Table 1 ijms-25-10976-t001:** Antimicrobials used for MIC assay.

Antimicrobial	*Nocardia cyriacigeorgica*—MIC (μg/mL)											Resistance Rate (%)	MIC_90_ (μg/mL)
	0.5	1	2	4	8	16	32	64	128	256	512		
Gentamycin				5	2			1	2		5	10/15 (66.7)	512
Enrofloxacin				1		1				13		15/15 (100)	256
Penicillin											15	15/15 (100)	512
Amoxicillin							2			13		15/15 (100)	256
cephalonium						1	2			12		15/15 (100)	256
Cefalexin											15	15/15 (100)	512
Ceftriaxone				1					2	12		15/15 (100)	256
Ceftiofur			1						2	12		14/15 (93.3)	256
Lincomycin											15	15/15 (100)	-
Erythromycin											14	15/15 (100)	-

**Table 2 ijms-25-10976-t002:** Primer sequences of bovine genes used for quantitative PCR.

Gene	Primer Sequence		Reference
TNF-α	Upstream	5′-ACGGGCTTTACCTCATCTACTC	Liu et al., 2020 [49]
	Downstream	3′-GCTCTTGATGGCAGACAGG	
IL-1β	Upstream	5′-AGGTGGTGTCGGTCATCGT	Liu et al., 2020 [49]
	Downstream	3′-GCTCTCTGTCCTGGAGTTTGC	
IL-6	Upstream	5′-ATCAGAACACTGATCCAGATCC	Liu et al., 2020 [49]
	Downstream	3′-CAAGGTTTCTCAGGATGAGG	
IL-8	Upstream	5′-ACACATTCCACACCTTTCCA	Liu et al., 2020 [49]
	Downstream	3′-GGTTTAGGCAGACCTCGTTT	
GAPDH	Upstream	5′-CATTGACCTTCACTACATGGT	Liu et al., 2020 [49]
	Downstream	3′-ACCCTTCAAGTGAGCCCCAG	

## Data Availability

The raw data supporting the conclusions of this manuscript will be made available by the authors, without undue reservation, to any qualified researcher.

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
