# Peer review of "Selenomethionine Mitigates Effects of Nocardia cyriacigeorgica-Induced Inflammation, Oxidative Stress, and Apoptosis in Bovine Mammary Epithelial Cells"

_ijms, 2024, doi:10.3390/ijms252010976_

Round 1
Reviewer 1 Report
Comments and Suggestions for Authors
This study investigates the protective effects of selenomethionine against N. cyriacigeorgica infection on bovine mammary epithelial cells in vitro. N. cyriacigeorgica-induced inflammatory cytokine regulation, oxidative stress and apoptosis are shown to be reduced by pre-treatment of SeMet. However, this manuscript is not easy to read, and the conditions of experiment arrangement as well as results and discussion are difficult to understand. This reviewer suggests a major revision of this manuscript is required with better organization and explanation of experiment designation and data interpretation.
Major comments
1. This manuscript is not written in a proper way of academic writing and requires a careful proof-reading by a professional editing service.
2. [Introduction]: The pathological features of N. cyriacigeorgica-induced mastitis on mammary epithelial cells in vivo shall be described carefully. N. cyriacigeorgica induced cell death may not only apoptosis, but also other types of cell death mechanism. This study's reason for investigating N. cyriacigeorgica-induced apoptosis alone must be clearly explained. Moreover, mastitis is a disease involves with various cell types in mammary tissues such mammary epithelial cells, stroma cells, immune cells. This study is limited to the in vitro model using mammary epithelial cells alone and may not represent the real responses of the mammary tissue against infection of N. cyriacigeorgica in vivo. Eventually, the infectious scenario of mastitis usually includes a large amount of cell necrosis rather than apoptosis. This limitation has to be clearly described in introduction section, and have to be carefully discussed in discussion section.
3. [Introduction] The main aim of this study is to investigate the protective effects of selenomethionine, as described in Line 72. However, Table 1 shows the MIC assay of antimicrobials on N. cyriacigeorgica without a proper explanation to clarify the reason for performing this experiment. This manuscript also does not discuss the relationship or probable interaction between SeMet and antimicrobials with their results.
4. [Results] The conditions of treatment duration between SeMet and N. cyriacigeorgica are different among all experiments. Please provide a table or figure to explain the selection of different treatment durations among all experiments clearly and carefully.
5. [Results] The cell viability LDH are presented as % of control, but the value is 0 to 2.5 % in Figure 1. Does it mean the most treated (98~99%) cells are dead? There shall be a blank group treating with no SeMet and N. cyriacigeorgica and it serves as control of 100%.
6. [Results] Please provide quantitative data to evaluate the morphological changes, for example, the swollen endoplasmic reticulum, observed in Figure 2.
7. [Discussion] Please discuss the limitations of this in vitro model using mammary epithelial cells alone compared to in vivo mastitis events in mammary tissue composed of various cell types. Moreover, the dose of 40 μM SeMet, which demonstrates observed effects, is much higher than serum levels of selenium
(less than 1 μM in serum). Please discuss if it is possible to achieve this concentration in vivo.
8. [Discussion] Line 238: Under acute infection, mammary epithelial cells have to express and release proper signals to trigger innate immune responses against pathogens. Pro-inflammatory cytokines are key factors to activate such immune responses, and disruption of pro-inflammatory cytokine expression and secretion may suppress the immune defenses during the actual infectious scenario. Please give evidence to explain why the SeMet-suppressed pro-inflammatory cytokines observed in this in vitro study are beneficial during the acute infection of N. cyriacigeorgica, but do not disrupt the innate immune responses against this infection event.
9. [Discussion] Line 259: According to Figure 7, SeMet treatment only provides a slight protective effect on N. cyriacigeorgica-induced apoptosis (less than 10% of apoptotic cells). This result eventually suggests that SeMet treatment alone is not effective in protecting mammary epithelial cells from N. cyriacigeorgica infection.
9. [Materials and methods]Line 282: If the bovine mammary epithelial cells were isolated from cows, the status of animals, and methodology of isolation and purification of cells have to be clearly described. If it is a purchased cell line, please indicate the original source.
10. [Materials and methods] Line 289: The identification and purity of isolated N.cyriagerogica shall be clearly described, particularly when it was isolated from mastitis cases.
Comments on the Quality of English LanguageThis manuscript is not written in the proper academic style and requires careful proofreading by a professional editing service.
Reviewer 2 Report
Comments and Suggestions for Authors
Dear Authors,
congratulations on very interesting manuscript. In my opinion it is well written, clearly and concisely, but all relevant details are presented.
Below are my insights:
Abstract – there are too many undescribed abbreviations. Either provide full names or try to shorten the fragments that need these abbreviations.
Introduction is comprehensive and provides all necessary information to understand the background of the undertaken experiments.
Results are clearly presented and described in SEM pictures, graphs and tables. The results are also explicitly explained and discussed with references to numerous studies of other Researchers.
The conclusions drawn from the study are not far-fetched, but promising.
The methodology is presented using a graph, table and clearly described in relevant subsections.
My only remark that is even not critical but perhaps worth considering is that there are many abbreviations throughout the text, which in some fragments make the substantial matter of the manuscript difficult to follow. I wonder if it is possible to reduce the number of abbreviations.
Round 2
Reviewer 1 Report
Comments and Suggestions for Authors
NA